# Measuring Quality of Life in Adults with Scoliosis: A Cross-Sectional Study Comparing SRS-22 and ISYQOL Questionnaires

**DOI:** 10.3390/jcm12155071

**Published:** 2023-08-01

**Authors:** Fabio Zaina, Irene Ferrario, Antonio Caronni, Stefano Scarano, Sabrina Donzelli, Stefano Negrini

**Affiliations:** 1ISICO (Italian Scientific Spine Institute), Via Roberto Bellarmino 13/1, 20141 Milan, Italy; 2IRCCS, Istituto Auxologico Italiano, Department of Neurorehabilitation Sciences, Ospedale San Luca, 20149 Milan, Italy; 3Department of Biomedical Sciences for Health, Università degli Studi di Milano, 20133 Milan, Italy; 4Department of Biomedical, Surgical and Dental Sciences, University “La Statale”, 20122 Milan, Italy; 5IRCCS Istituto Ortopedico Galeazzi, 20157 Milan, Italy

**Keywords:** quality of life, adult scoliosis, rasch analysis, psychometrics

## Abstract

Idiopathic scoliosis is common in adulthood and can impact patients’ physical and psychological health. The Scoliosis Research Society-22 Questionnaire (SRS-22) has been designed to assess health-related quality of life (HRQOL) in idiopathic scoliosis, and it is the most used disease-specific outcome tool from adolescence to adulthood. More recently, the Italian Spine Youth Quality of Life (ISYQOL) international questionnaire was developed, which performs better than SRS-22 in adolescent spinal deformities. However, the ISYQOL questionnaire has never been tested in adults. This study compares the construct validity of ISYQOL and SRS-22 with the Rasch analysis (partial credit model). We recruited 150 adults and 50 adolescents with scoliosis (≥30° Cobb). SRS-22, but not ISQYOL, showed disordered categories and one item not fitting the Rasch model. A 21-item SRS-22 version with revised categories was arranged and further compared to ISYQOL. Both questionnaires showed multidimensionality, and some items (SRS-22 in a greater number) functioned differently in persons of different ages. However, the artefacts caused by multidimensionality and differential functioning had a low impact on the questionnaires’ measures. The construct validity of ISYQOL International and the revised SRS-22 are comparable. Both questionnaires (but not the original SRS-22) can return measures of disease burden in adults with scoliosis.

## 1. Introduction

Spinal deformities, such as scoliosis, may significantly impact patients’ physical and psychological health [1]. Adolescents with idiopathic scoliosis can show psychological and emotional distress, with anxiety as the most common symptom [2]. They may exhibit poorer psychosocial functioning and body image than their healthy peers, while adults with scoliosis show concerns about the risk of disability, body image, and physical health problems [3]. During adulthood, this pathology can cause lower back pain, bent posture, shortness of breath, and reduced autonomy in everyday activities [4]. Disease-specific outcome tools can assess the extent of this impact, e.g., the Scoliosis Research Society-22 Questionnaire (SRS-22), the most used instrument to assess health-related quality of life (HRQOL) in patients with idiopathic scoliosis [5]. Initially developed [5,6] in a young population, many studies have examined its application for adult spinal deformities, demonstrating its usefulness in this population [7,8,9]. Nevertheless, other papers showed drawbacks and limitations [10].

When used as an HRQOL measure, the SRS-22, developed in the classical test theory framework (CTT), the oldest set of psychometrics techniques for developing scales and questionnaires, has a significant flaw: its total ordinal score is not a measure but a measure approximation at best [11]. Equal changes in ordinal scores do not necessarily reflect identical changes in the quantity of the variable of interest. This fact has practical consequences: customary statistics such as effect size can be misleading when calculated on ordinal scores.

Like CTT, the Rasch analysis is a statistical method designed to build and assess questionnaires. If a questionnaire’s score empirically demonstrates compliance with the assumptions of the Rasch analysis, it is possible to turn these total scores into actual interval measures [11].

Rasch’s analysis revealed that the SRS-22 has poor metric properties, failing to assess HRQOL properly in non-surgical adolescents and children [12]. Therefore, we developed the Italian Spine Youth Quality of Life (ISYQOL), using Rasch analysis, as a new patient-reported outcome measure to assess HRQOL in adolescents with spinal deformities [13]. ISYQOL had satisfactory construct validity and, compared to SRS-22, better known-groups validity, detecting the impact of disease severity on HRQOL [14]. More recently, a different version called “ISYQOL International” has been validated in a multicentre international study, the cross-culturally equivalent version of the questionnaire [15].

To our knowledge, no other Rasch-consistent questionnaire measuring HRQOL in adults with spinal deformities is available, and the data on ISYQOL’s and ISYQOL International’s validity come solely from the adolescent population. Therefore, the present study aims to verify the construct validity of ISYQOL International and to compare its properties to the SRS-22 in adults with scoliosis. We hypothesize that the ISYQOL can perform at least as well as the SRS-22 in adults with scoliosis. Moreover, we expect ISYQOL to perform similarly in adults and adolescents with scoliosis.

## 2. Materials and Methods

### 2.1. Study Characteristics

We conducted a cross-sectional study based on data from an ongoing prospective database collecting records from patients attending a tertiary outpatient clinic specializing in the conservative treatment of spinal deformities in Italy.

### 2.2. Data Collection

As standard practice, all patients attending our rehabilitation centre complete the self-administered SRS-22 and ISYQOL questionnaires before every medical consultation.

### 2.3. Participants

On 8 October 2022, we extracted all consecutive patients respecting the following criteria: (1) age ≥18 years, (2) diagnosis of idiopathic scoliosis with a curve of 30° Cobb or more, and (3) availability of both the ISYQOL and SRS-22 questionnaires. Exclusion criteria were the following: (1) history of spine surgery, (2) history of relevant diseases, surgery, or trauma, and (3) a positive neurologic examination. Only questionnaires not exceeding two missing answers were included in the analysis. From this group, we randomly extracted 150 patients. Since we expected that age could impact the results of the questionnaires, we made a cluster sampling based on age and sex. We had six groups based on age: 20–29, 30–39, 40–49, 50–59, 60–69, and 70–79 years. For each group, we extracted 20 females and five males as per the different sex prevalence of spinal deformities. This is based on the published literature and our data. A systematic review has reported a prevalence of degenerative scoliosis of 41.2% for females versus 27.5% for males [16]. Considering idiopathic scoliosis, the ratio is 7/1 in favour of females [1]. In our database, which includes a mixed population, the ratio is about 4–5/1 for all kinds of scoliosis during adulthood.

Moreover, we randomly extracted a sample of 50 individuals aged between 14 and 18 years from the dataset we analysed in our previous study for the ISYQOL validation study [14]. We included ten participants for each of the five years of age (eight females and two males), all affected by idiopathic scoliosis and all not wearing a brace.

### 2.4. Sample Size Calculation

In the Rasch analysis framework, about 200 questionnaires are usually enough to produce stable estimates [17]. In addition, we arranged age subgroups of equal size to comply with some recent guidelines and recommendations for the differential item functioning (DIF) analysis (see below) [18,19].

### 2.5. Health-Related Quality-of-Life Questionnaires: SRS-22 and ISYQOL

The SRS-22 questionnaire [5] consists of 22 items scored on five ordinal categories (1–5), with higher scores corresponding to a lower disease burden and, thus, better quality of life. It measures specific aspects of HRQOL, covering five domains: self-image, mental health, pain, function, and treatment satisfaction [20,21].

ISYQOL International derives from ISYQOL. We translated ISYQOL into different languages and assessed its cross-cultural validity. We removed four items from the original questionnaire [15]. The ISYQOL International consists of 16 items scored on three categories (0–2); the higher the category numeral, the more the disease burden. The ordinal ISYQOL total score is converted into an interval measure with logit as the measurement unit (the higher the logit measure, the higher the disease burden). The ISYQOL ordinal score can also be expressed on an interval scale ranging from 0 to 100%, with 100% indicating an excellent quality of life. It consists of two subscales, one (9 items) regarding spine health and the other (7 items) regarding brace wearing. Only the ISYQOL International spine domain was collected here since no participant wore a brace at the point of inclusion in the study.

### 2.6. Statistical Analysis

We ran the Rasch analysis in the following steps [12,13,15] (Appendix A and Appendix B).

#### 2.6.1. Categories’ Functioning

The categories’ functioning was evaluated by assessing their average order, as per Linacre [22], and the order of the modal thresholds, as per Andrich [23].

#### 2.6.2. Fit the Model

We can extract measures from the questionnaire’s scores if categories are ordered and data fit one of the Rasch family models (here, Masters’ partial credit model [24]). 

We used the mean square (MnSq) and the z-standardised (Z-Std) statistics (“infit” and “outfit” variants) to quantify the departure of the observed data from the model’s prediction and the probability that this departure was due to chance, respectively. 

Here, an item was considered to “misfit”, i.e., not fit the model adequately, if:

outfit MnSq > 2.0 and absolute outfit Z-Std > 1.96, or 

infit MnSq > 1.5 and absolute infit Z-Std > 1.96.

#### 2.6.3. Dimensionality

Measures are unidimensional, meaning they reflect a single variable’s amount. In the Rasch framework, principal components with an eigenvalue >2 from a principal component analysis (PCA) calculated on the model’s residuals indicate multidimensionality.

In the case that multidimensionality is found, whether this multidimensionality harms measurements can be tested by assessing if cluster 1 items (items with a large and positive loading on the principal component) and cluster 3 items (items with a large and negative loading) return a different participants measure from cluster 2 items (those items loading low on the principal component, thus reflecting only the variable grasped by the model of Rasch).

Suppose persons’ measures from cluster 1 and cluster 3 are comparable. In that case, the artefact caused by the hidden variable highlighted from the principal component is not strong enough to cause a severe measurement artefact [25]. For this comparison, we used ANOVA.

#### 2.6.4. Differential Item Functioning

Differential item functioning (DIF) indicates that an item does not work the same in different groups of respondents. Given the study’s aim, the current analysis focused on the DIF for age. We reorganized the participants’ sample into the following age classes: adolescents (from 14 to 18 years), young adults (from 20 to 39 years), middle-aged adults (from 40 to 59 years) and older adults (from 60 to 79 years). As a complementary analysis, we evaluated DIF for gender (males vs. females). We tested the DIF of SRS-22 and ISYQOL International items following Linacre [25].

Suppose the calibration of an item is different in a subgroup of participants and in the primary analysis. If this difference is <0.5 logit with *p* > 0.01, the DIF can be considered too small to matter.

In the case of a large (>0.5 logit) and significant (*p* < 0.01) DIF being found for an item, the observed scores of the participants’ subset on this item and their expected scores are compared to provide an easy understanding of the artefact caused by the DIF in terms of the questionnaire’s total score.

#### 2.6.5. Questionnaire Reliability and Targeting

We reported the ISYQOL International and the SRS-22 reliability as “Rasch persons’ reliability”, similar to Cronbach’s alpha. From this reliability index, we calculated the number of strata, the number of significantly different levels of the disease burden a person can progress through (Supplementary Materials 1 in [26]).

Floor and ceiling effects were calculated as the percentage of respondents obtaining the minimum and maximum total questionnaire scores, respectively. The size of the difference between the persons and the items measures complements this information. A questionnaire with no floor effect, no ceiling effect, and 0 logit difference between participants and items mean measure targets appropriately the recruited sample participants. The item and person maps graphically show the targeting of persons compared to the measurement instrument.

Finally, we provide the score-to-measure tables to allow future users to turn the questionnaires’ total scores into interval measures.

We used FACETS 3.84.0 and WINSTEPS 5.4.3.0 for the Rasch analysis (partial credit model). We performed the statistics using the R (R version 4.2.3 “Shortstop Beagle”) software. Type 1 error probability was set to 0.05 as customary in all analyses, but we lowered this threshold for DIF to 0.01 because of multiple statistical testing [15,27].

### 2.7. Ethical Approval

The local ethics committee approved the study (Comitato Etico Milano Area 2, 215_2022bis), and we registered the protocol on clinicaltrials.gov (NCT05333757). This study did not receive dedicated funding support. All participants gave written informed consent.

## 3. Results

At the time of data extraction, our database included 3254 adult patients with scoliosis or other spinal deformities (2540 females, 714 males), fulfilling the inclusion criteria. From these, we randomly selected 150 subjects (120 females, 30 males). For each group, we had 20 females and five males diagnosed with scoliosis based on clinical and radiological assessment.

Table 1 reports the clinical features of the participants included in the current analysis.

### 3.1. Rasch Analysis of ISYQOL International

All nine items of ISYQOL International had ordered categories and thresholds (Table A1 in Appendix B). 

Regarding the fit to the model, infit and outfit MnSq were suitable for all the questionnaire’s items (Table 2).

The PCA of the model’s residuals highlighted that another dimension, in addition to the one taken into account by the Rasch model, affects the questionnaire scores. The eigenvalue of the first principal component was 2.55, a value which indicates that the hidden dimension affects the score of three items at most.

Cluster 1, i.e., the cluster of items with a positive loading on the first principal component, included items 6, 8, and 9 (8, 11, and 12 of ISYQOL original; Figure 1). Notably, all these three items had a large (>0.60) loading. Cluster 3 (i.e., the items with negative loading) included items 1, 2, and 7 (1, 2, and 9 of ISYQOL original).

ANOVA comparing the persons’ measures from cluster 1, 2, and 3 items was not significant (F_2,368_ = 1.09; *p* = 0.337), indicating that on average, clusters 1 and 3, i.e., the clusters of items more severely affected by the first principal component hidden variable, measure the same as cluster 2 items, i.e., the items prominently reflecting the variable grasped by the model of Rasch. 

Table 3 reports the results of the DIF analysis.

One item only (item 8, corresponding to item 11 in the ISYQOL original) was affected by DIF for age.

In detail, item 8’s calibration was lower in adolescents than in the primary analysis, including participants of all ages (calibration difference = 0.83 logits, *p* = 0.006).

The age-related DIF for item 8 indicates that adolescents are more likely to be bothered than young, middle-aged, and older people by showing their physical appearance despite the same overall burden of disease level.

Even if large at the item level and statistically significant, the age-related DIF of item 8 caused a minor artefact on the ISYQOL total score (and hence on the ISYQOL measures). On average, adolescents scored more than expected on item 8. However, the difference between the observed score on item 8 (biased since inflated by DIF) and the expected score given the primary analysis was 0.19 (i.e., less than one-fifth of a point of the ISYQOL International total score).

We found no DIF for gender.

The ISYQOL International’s reliability (model, sample reliability, extremes included) was 0.88, which allows for distinguishing 3.91 strata. The questionnaire targeting was satisfactory, as indicated by a participant’s mean measure of 0.27 logits (SD = 2.52 logits).

Regarding the ceiling and floor effect, ten participants (out of 200, i.e., 5%) obtained the maximum score and five (i.e., 2.5%) the minimum. 

Figure 2 shows the item and person maps of ISYQOL International. Table 4 provides the ISYQOL International score-to-measure conversion table.

### 3.2. Rasch Analysis of SRS-22

On the first analysis run, 11 items had disordered categories. One possible reason was that the respondents seldom selected the lower categories. As a result, the accuracy of estimating the categories’ parameters was poor.

We rearranged items 7, 8, 9, 13, 17, and 20 by collapsing the original categories 1 and 2 into the new category 1. For items 5, 11, 15, 18, and 22, it was necessary to collapse categories 1, 2, and 3. Note that after this procedure, SRS-22 consisted of a mixture of items scored on five (11 items), four (6 items), and three (5 items) categories.

The collapsing procedure efficiently ordered all the items’ categories (Table A2 in Appendix B) 

However, modal thresholds were disordered in seven items (7, 9, 12, 15, 16, 17, and 19). 

On a subsequent analysis run, item 15 did not fit the model because of a large and significant outfit (MnSq = 2.97; Z-Std = 3.30). On a new run in which item 15 was dropped from the questionnaire, all 21 items properly fit the model (Table 5). The analysis continues assessing the measurement properties of this revised version of the SRS-22.

The PCA of the residuals highlighted two hidden dimensions, as indicated by the eigenvalue of the first principal component (3.45) and that of the second (2.72).

Items 1, 2, and 12 were the three items with the largest loading of cluster 1 (Figure 3). Items 4, 10, and 19 were the three with the largest negative loading (i.e., cluster 3 items with the largest loading). Regarding the second principal component, the three cluster 1 items were items 7, 13, and 16. The three most significant cluster 3 items were items 10, 19, and 21. 

Despite the presence of two additional dimensions, person measures from the three items clusters were not significantly different from each other (contrasts on the first principal component: F_2,396_ = 1.57; *p* = 0.209; contrasts on the second principal component: F_2,396_ = 0.60; *p* = 0.549). 

Four items (i.e., items 3, 4, 8, and 12) were affected by DIF for age (Table 3).

Item 3’s calibration was significantly lower when calculated in the older adults group than when we inputted the total participants’ sample into the analysis. (i.e., item 3’s calibration was lower in older persons than in middle-aged, young adults, and adolescents). We found the same pattern for item 8. Item 4’s calibration was higher, and item 12’s was lower in adolescents.

Due to these differences in the items’ calibrations, the older adults observed scores on items 3 and 8 was larger than expected. Adolescents’ scores on item 4 were lower than predicted, while on item 12 were higher.

However, when we consider the artefact they cause in the SRS-22 total score, the biases of items 4 and 12 have opposite signs (the first decreases and the second increases the item’s score), thus compensating each other. The bias of items 3 and 8 inflates the SRS-22 total score by 0.31 and 0.36 points (i.e., 0.67 points when considered together) in older persons. Similarly to ISYQOL International, DIF is present, but its consequences on the measures derived from the total questionnaire score seem modest. 

We found no DIF for gender. 

The reliability of the modified version of the SRS-22 questionnaire was 0.91, and the number of strata was 4.59.

Only one respondent obtained the SRS-22 maximum score. However, the participants’ mean measure was 0.86 logits (SD = 1.18 logits), flagging poor targeting of the SRS-22 questionnaire in this sample (Figure 4).

We provide the score-to-measure table of the revised SRS-22 version in Table 6.

## 4. Discussion

Spinal deformities can negatively impact a patient’s quality of life during adulthood. To monitor the changes over time, clinicians need specific tools to picture the patient’s pain, activity limitations, and participation restrictions. Many validated and reliable tools are available for patients with chronic LBP [28]. They can also help in the case of spinal deformities but could lack some specificity. From a psychometrics perspective, their content validity is poor. For example, some items included in the Oswestry Disability Index (ODI), such as rest quality and travelling, are not specific for spinal deformities. In a recent study about bracing, despite the significant improvements in pain, the ODI failed to show clinically significant improvements [29]. In a sample of surgically treated patients, the SRS self-image domain demonstrated higher responsiveness to change, followed by SRS total, then SRS pain, and then ODI [7]. Unfortunately, it is unclear whether it was a limit of the ODI, or an issue related to the too-small clinical changes of patients. These findings and limits suggest the need for developing specific tools. The SRS-22 was explicitly designed for adolescent scoliosis patients managed in a surgical setting. For those treated conservatively, they showed some limits and mainly a ceiling effect [12]. Many authors and clinicians use the SRS-22 also in adults even though young patients were its original target, and some limits have already been reported [10]. The SRS-22 remains the most widely used questionnaire in adults with spinal deformities. Nevertheless, the challenges with the currently accepted standard questionnaire (SRS-22) for HRQOL assessments in scoliosis are detailed in the literature and application of the SRS-22 in the adult population with scoliotic deformities has been debated [30]. Currently, there is no gold standard that is reliable and valid for the complexity of the ‘patient’s perception’ on how their deformity impacts their life. Recently, we developed a new tool, the ISYQOL, to measure conservatively managed patients during growth appropriately, but no data are available for adults. The current one was the first study to compare the properties of the ISYQOL to the SRS-22 in adults attending a rehabilitation centre specialized in the conservative treatment of spinal deformities.

Regarding the Rasch analysis, the original SRS-22 questionnaire, but not ISYQOL International, failed to meet the two basic assumptions of the analysis: the assumption of ordered categories and data-model fit.

Several SRS-22 items had disordered categories and thresholds, and disordered thresholds remain even after rearranging the categories so that their average measure is ordered. In addition, item 15 of SRS-22 does not fit the model. Therefore, in the fundamental measurement framework [11,31], the SRS-22 should not be used in its original form to measure the disease burden in adults with spinal deformities.

Despite rearranging the SRS-22 to comply with the ordered categories and data-model fit assumptions, multidimensionality still affects it, and DIF corrupts several items for age and gender. ISYQOL International suffers similar issues in this respect. However, regarding multidimensionality, SRS-22’s measures of HRQOL are disturbed by two additional unknown variables, while those from ISYQOL International are disturbed by one. The SRS-22 is tridimensional, while ISYQOL International is bidimensional: considering that accurate measures are unidimensional [11], we can assume the latter to be better than the former.

Regarding DIF, DIF for age afflicts more SRS-22 than ISYQOL items.

From a measurement theory perspective, multidimensionality and DIF are serious flaws. However, the total questionnaire score and the measures extracted with the Rasch analysis from these scores are robust to some DIF and multidimensionality [32]. If a questionnaire demonstrates this measure’s robustness, we can safely use it despite these flaws. Based on our findings, the ISYQOL International and the modified SRS-22 version can measure the disease burden despite the DIF and multidimensionality, since we experimentally found these flaws are negligible. However, the artefacts caused by DIF and multidimensionality would likely be non-negligible if single or groups of items were selected from the questionnaire and used for measuring, a frequently used practice for SRS-22 [7].

ISYQOL International has two additional strengths: it is shorter and more straightforward than the SRS-22 and better targeted than SRS-22. About this last point, the average SRS-22 measure is larger than 0 logits, indicating that several SRS-22 items investigate a (low) range of HRQOL, which the patients included here do not experience. SRS-22 is not perfectly tuned to measure patients like those recruited here.

On the contrary, SRS-22’s reliability is better than that of ISYQOL International, a finding which results from its large number of categories times items. However, the modest improvement in the reliability of SRS-22 comes at the expense of a more marked increase in the number of categories and items (91 for SRS-22 and 27 for the ISQYOL International—spine domain).

We already assessed the measurement properties of the SRS-22 with the Rasch analysis [12], and our previous study also pointed out different problems. However, in the current work, a more liberal analysis has been conducted, so the SRS-22 flaws seem less severe. Nevertheless, even if adherence to the analysis requirements is relaxed as much as possible, some significant drawbacks remain, such as disordered categories and a misfitting item.

Another reason for the different results of the current and our former work is that the participants recruited here were mostly adults. At the same time, previously, we studied SRS-22 functioning in children and adolescents. The DIF analysis highlights that, in most cases, adolescents usually understand several SRS-22 items differently from adults. Hence, SRS-22 could function differently in young people than adults, but further research is needed.

### Study Limitations and Further Developments

The SRS-22 and ISYQOL International questionnaires demonstrated multidimensionality, suggesting they measure multiple HRQOL aspects. It has been empirically shown here that this multidimensionality is unlikely to harm. However, multidimensionality is always a measurement threat strictly, making the questionnaires’ interpretation more challenging. In this regard, the additional hidden variable in ISYQOL International’s scores and the two hidden variables in the SRS-22 remain to be discovered.

The same reasoning applies to the results of the DIF analysis (to note, DIF is simply another form of multidimensionality). The study found that some questionnaire items functioned differently in individuals of different ages. Furthermore, in this case it is shown that the measurement artefact caused by DIF is negligible. However, in strict metrological terms, this response bias indicates that the questionnaires do not perform consistently across different age groups.

ISYQOL is a relatively new instrument, and studies are needed to assess it further. Recently, ISYQOL has been translated into different languages and tested in different cultures in young persons with scoliosis [15]. There is a need to compare ISYQOL International and SRS-22 in adult patients from different countries and cultures as well. We could also test ISYQOL’s properties in patients who underwent spine surgery and compare it to other quality-of-life measures in addition to SRS-22. Finally, ISYQOL International has no items assessing pain, which can be a significant complaint adults make [3]. If this is an issue regarding ISQOL International’s face validity when used to evaluate the scoliosis burden of disease in adults, it remains to be investigated.

## 5. Conclusions

Scoliosis treatment cannot be restricted solely to correcting the curvature, but it should also assess and monitor patients’ satisfaction, psychological issues, and HRQOL over time. There is a need for a proper tool that allows clinicians to evaluate the impact of spinal deformities in adulthood. The results of the present work indicate that the ISYQOL spine health subscale can be administered in a clinical setting to evaluate HRQOL in adults with scoliosis. SRS-22, in its original form, showed poor construct validity in the Rasch analysis measurement framework. While the revised SRS-22 has improved metrological features, ISYQOL International is better regarding dimensionality and differential item functioning. In addition, ISYQOL International is also considerably shorter, more straightforward, and better targeted to measure the disease burden in adults with non-surgical scoliosis.

## Figures and Tables

**Figure 1 jcm-12-05071-f001:**
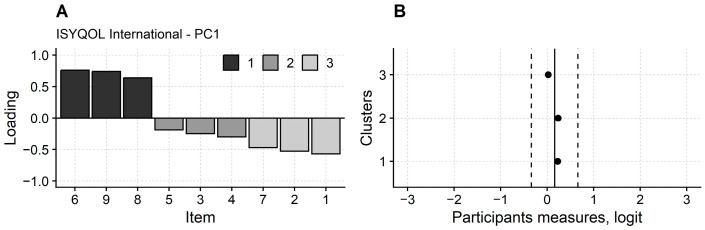
ISYQOL International dimensionality. The principal component analysis (PCA) calculated on the model’s residuals is provided for the ISYQOL International questionnaire. Panel (**A**): loadings of the ISYQOL International items on the first principal component from the PCA. The items are grouped into three clusters (cluster 1, 2, and 3). Cluster 2 (dark grey) items load low in absolute value on the principal component. Their score is scarcely affected by the hidden variable flagged by this component but mainly reflects the variable, i.e., disease burden, grasped by the Rasch model. On the contrary, the score of cluster 1 items (black) is inflated by an additional hidden variable, while that of cluster 3 items (light grey) is reduced. Panel (**B**): participants are measured with cluster 1, 2, and 3 items, and their mean measure is compared (black dots). Vertical continuous line: participants’ mean measures from the total ISYQOL International. Vertical dashed lines: participants’ mean measures from the total ISYQOL International ± 0.5 logit. On average, the participants’ measures from the three clusters of items are only slightly different from each other and minimally different from the participants’ measures from the full ISYQOL International. In particular, the mean difference between the clusters and the total questionnaire measures is well below 0.5 logits. Even if an additional unwanted variable contaminates the scores of cluster 1 and 3 items, this variable causes a negligible measurement artefact. Extreme persons, i.e., those obtaining the maximum or minimum questionnaire total score, whose real measure is unknown, have not been considered in this analysis.

**Figure 2 jcm-12-05071-f002:**
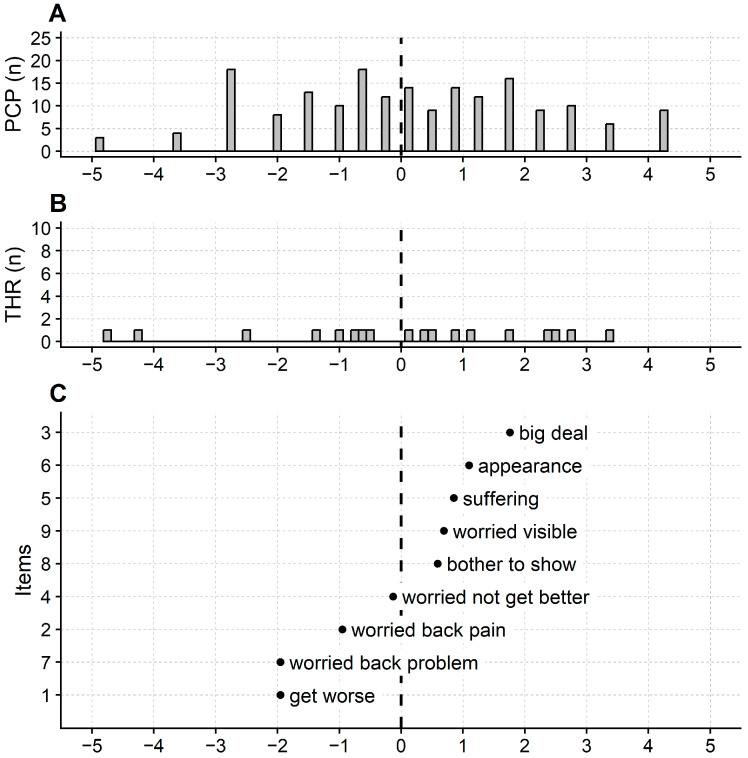
ISYQOL International maps. Maps of persons (**A**), thresholds (**B**), and items (**C**) of the ISYQOL International questionnaire (spine domain). PCP: participants; n: number of; THR: Andrich’s thresholds. X-axis: line of the construct (i.e., the disease burden continuum) measured in logits. The disease burden increases from left to right. ISYQOL logit measures are measures of disease burden: the higher the logit measure, the more the disease burden. Rightmost persons on the disease burden line (**A**) suffer a high disease burden. The rightmost items (**C**) flag a high disease burden: only persons suffering a high disease burden will affirm the content of these items. In (**C**), the Y-axis reports the ISYQOL International item number. Labels in plot (**C**) are keywords recollecting the item content. The dot position on the X-axis returns the item measures, called here “item calibration”. Vertical dashed segment: items mean calibration, set to 0 logits, as customary.

**Figure 3 jcm-12-05071-f003:**
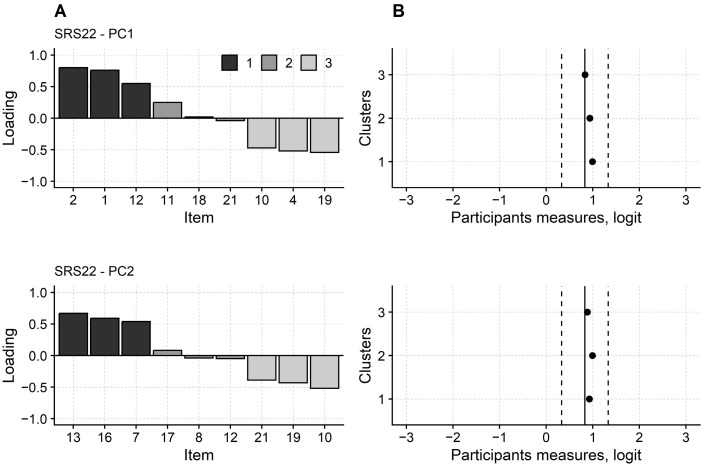
SRS-22 dimensionality. The principal component analysis (PCA) of the SRS-22 questionnaire (revised version) highlighted two principal components, indicating that the score of some SRS-22 items were affected by two hidden variables in addition to the Rasch dimension. Same symbols and abbreviations as Figure 1. Upper graphs in panels (**A**,**B**) report the analysis for the first principal component. The lower graphs report the clusters on the second principal component. The revised SRS-22 consists of 21 items. Here, are only the three cluster 1 items with the largest positive loadings, the three cluster 3 items with the largest negative loadings, and the three cluster 2 items with the most negligible loading.

**Figure 4 jcm-12-05071-f004:**
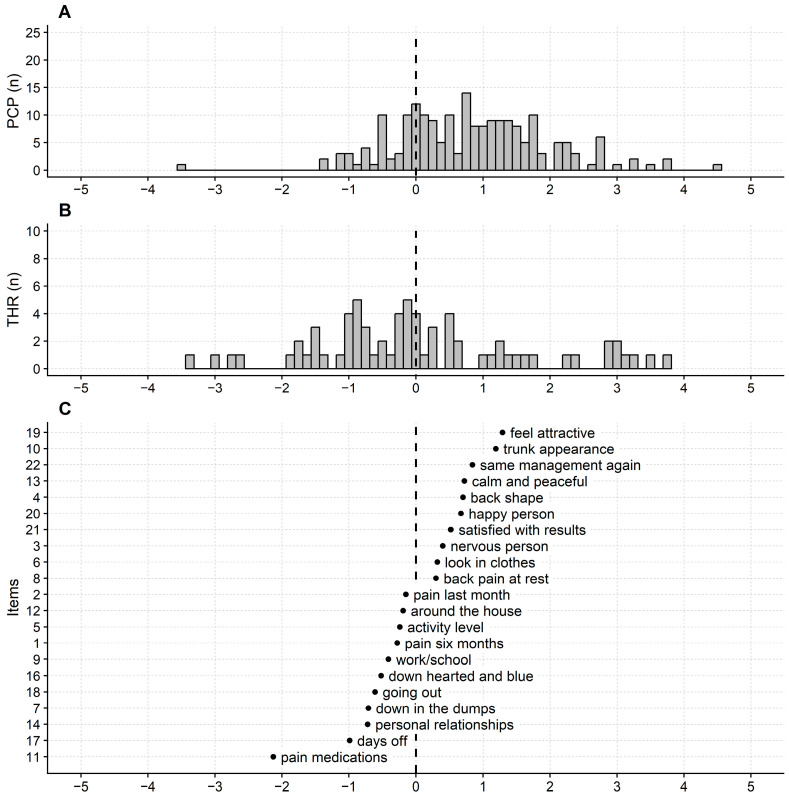
Revised SRS-22 maps. Same abbreviations as Figure 1. SRS-22 logit measures are measures of quality of life. So, the higher the logit measure, the higher the quality of life. People experiencing a full quality of life are on the right of the continuum, and quality of life decreases from right to left. Note that the persons map (**A**) histogram is displaced to the right (e.g., the distribution mode is about 0.75 logits). This indicates that persons score high on the questionnaire and that the SRS-22 items are too easy to endorse for the participants’ sample recruited here. The thresholds histogram (**B**) shows several thresholds with overlapping calibrations between −1 and 0 logits. While many thresholds (or items) within the same construct range increase the measurement precision, it also points out some redundancy in the questionnaire. (**C**) items map.

**Table 1 jcm-12-05071-t001:** Participants’ clinical data.

	Adults	Adolescents
Males vs. females, N	30 vs. 120	10 vs. 40
Mean age (SD), years	49 (17.8)	16 (1.4)
Mean disease severity (SD), °Cobb	46.2 (16.6)	24.7 (9)
Median TRACE score (IQR)	7 (4)	5 (3)

N: number of participants; SD: standard deviation; °Cobb: angle of scoliosis curvature measured according to Cobb; IQR: interquartile range; TRACE: trunk aesthetic clinical evaluation (ordinal score of back aesthetics ranging from 1 to 12, with a high score marking a poor trunk aesthetic appearance).

**Table 2 jcm-12-05071-t002:** Calibration of items of ISYQOL International and their fit to the model.

Items	Calibration	SE	Infit		Outfit	
			MnSq	Z-Std	MnSq	Z-Std
1 (1), get worse	−1.95	0.17	0.97	−0.30	0.99	0.00
2 (2), worried back pain	−0.95	0.15	1.12	1.17	1.54	3.11
3 (3), big deal	1.76	0.16	1.00	0.00	0.95	−0.23
4 (4), worried not get better	−0.13	0.14	1.01	0.10	1.08	0.61
5 (7), suffering	0.85	0.15	0.93	−0.64	0.84	−1.22
6 (8), appearance	1.10	0.15	0.82	−1.92	0.75	−1.81
7 (9), worried back problem	−1.95	0.18	0.83	−1.68	0.75	−1.72
8 (11), bother to show	0.59	0.14	1.25	2.31	1.24	1.53
9 (12), worried visible	0.69	0.15	1.07	0.69	1.02	0.23

Items: the item number and a keyword summarising the item content; the item number of ISYQOL original is also reported in brackets. Calibration: item calibration (i.e., item measure) expressed in logit. SE: standard error in the item calibration (logit). Infit: inlier sensitive fit indices; outfit: outlier sensitive fit indices. MnSq: mean square statistic; Z-std: z-standardised statistic.

**Table 3 jcm-12-05071-t003:** Age-related differential item functioning of the ISYQOL International and SRS-22 questionnaires.

Item	Group	Obs − Exp	Bias	SE	*p* Value
*ISYQOL International*					
8 (11), bother to show	Adolescents	0.19	0.83	0.28	0.006
*SRS-22*					
3, nervous person	Older	0.31	0.61	0.20	0.003
4, back shape	Adolescents	−0.46	0.68	0.17	0.000
8, back pain at rest	Older	0.36	0.51	0.17	0.004
12, around the house	Adolescents	0.39	1.19	0.35	0.002

Item: item number and its keyword; the item number of ISYQOL original is also provided in brackets. Group: participants group for which the item’s calibration differs from the primary analysis (e.g., the calibration of ISYQOL International item 8 is different in adolescents than in the primary analysis). Obs-Exp: artefact in the item score caused by differential item functioning (DIF) and expressed as the difference between the observed (Obs) and expected (Exp) score. The expected score is calculated given the item’s calibration from the primary analysis. For example, DIF for age inflates by 0.19 the score of adolescents on ISYQOL International item 8 (i.e., their score on this item is 0.19 points higher than it should be). Bias: absolute value difference, expressed in logits, between the item’s calibration from the primary analysis and the participants’ group. SE: standard error (logit) of the bias. *p* value: type 1 error probability of the t-test with the null hypothesis: “item’s calibrations in group and primary analysis are not different from each other”. For both the ISYQOL International (upper row) and the revised SRS-22 (lower rows), only the items with DIF > 0.5 logit with *p* < 0.01 are reported. No DIF was found for gender.

**Table 4 jcm-12-05071-t004:** Score-to-measure conversion table of ISYQOL International.

Score	Burden of Disease		HRQOL	
	Measure, Logit	SE, Logit	Measure, %	SE, %
0	−6.44	1.95	100.0	16.2
1	−4.89	1.24	87.1	10.3
2	−3.64	1.03	76.7	8.6
3	−2.73	0.89	69.1	7.4
4	−2.04	0.78	63.3	6.5
5	−1.49	0.71	58.8	5.9
6	−1.03	0.66	54.9	5.5
7	−0.61	0.63	51.4	5.3
8	−0.22	0.62	48.1	5.2
9	0.16	0.61	45.0	5.1
10	0.54	0.61	41.9	5.1
11	0.92	0.62	38.7	5.2
12	1.32	0.64	35.4	5.3
13	1.73	0.66	31.9	5.5
14	2.19	0.69	28.1	5.8
15	2.70	0.74	23.9	6.2
16	3.32	0.84	18.7	7.0
17	4.22	1.10	11.2	9.1
18	5.56	1.88	0.0	15.7

Score: ISYQOL International (spine health domain) total ordinal score. Measure, logit: interval measures of disease burden expressed in logits, i.e., the measurement unit of the Rasch analysis. Measure, %: interval measures reported on a user-friendly scale ranging from 0 to 100%, with 100% indicating full health-related quality of life (i.e., no disease burden). SE: standard error. Note that the higher the ISYQOL International total score, the higher the problems caused by the back condition to the patient (i.e., the higher the disease burden). The relationship between logit measures and ordinal scores is monotonic. Therefore, the higher the logit measure, the more the disease burden. Originally, ISYQOL was conceptualized as an HRQOL measure rather than a disease burden measure. For this reason, measure %, which is reversed compared to the total score and the logit measure, is also reported.

**Table 5 jcm-12-05071-t005:** Calibration of items of the revised SRS-22 and their fit to the model.

Items	Calibration	SE	Infit		Outfit	
			MnSq	Z-Std	MnSq	Z-Std
1, pain six months	−0.28	0.09	0.69	−3.72	0.66	−3.67
2, pain last month	−0.15	0.09	0.75	−2.83	0.71	−3.06
3, nervous person	0.40	0.10	1.14	1.41	1.21	2.02
4, back shape	0.70	0.09	1.09	0.96	1.10	1.00
5, activity level	−0.24	0.12	0.87	−1.46	0.85	−1.13
6, look in clothes	0.32	0.10	1.04	0.41	1.04	0.45
7, down in the dumps	−0.71	0.10	0.97	−0.27	1.13	0.68
8, back pain at rest	0.30	0.09	1.42	4.07	1.52	3.90
9, work/school	−0.41	0.09	1.09	0.79	1.08	0.44
10, trunk appearance	1.19	0.10	1.00	0.03	1.01	0.10
11, pain medications	−2.13	0.18	0.96	−0.24	1.00	0.06
12, around the house	−0.19	0.08	0.74	−2.73	0.81	−1.30
13, calm and peaceful	0.72	0.11	1.17	1.73	1.18	1.72
14, personal relationships	−0.72	0.09	0.80	−1.82	0.72	−1.63
16, downhearted and blue	−0.52	0.09	1.09	0.83	1.08	0.64
17, days off	−0.99	0.12	1.35	1.67	1.80	1.21
18, going out	−0.61	0.13	0.89	−1.10	0.98	−0.07
19, feel attractive	1.29	0.09	0.97	−0.22	0.91	−0.84
20, happy person	0.67	0.11	1.03	0.37	1.05	0.48
21, satisfied with results	0.52	0.09	0.95	−0.52	0.93	−0.65
22, same management again	0.84	0.11	1.20	2.26	1.24	2.01

Same abbreviations as Table 2. SRS-22 item 15 is not reported because of outfit values beyond the tolerance limits. The item numbering of the original SRS-22 was kept. Remember that the categories of the original SRS-22 items have been extensively rearranged.

**Table 6 jcm-12-05071-t006:** Score-to-measure conversion table of the revised SRS-22.

Score	HRQOL		HRQOL	
	Measure, Logit	SE, Logit	Measure, %	SE, %
21	−5.99	1.85	0.0	14.7
22	−4.73	1.04	10.1	8.3
23	−3.96	0.75	16.2	6.0
24	−3.49	0.63	19.9	5.0
25	−3.15	0.55	22.7	4.4
26	−2.88	0.50	24.8	4.0
27	−2.65	0.46	26.6	3.6
28	−2.46	0.42	28.2	3.4
29	−2.29	0.40	29.5	3.2
30	−2.14	0.37	30.7	3.0
31	−2.01	0.36	31.7	2.8
32	−1.89	0.34	32.7	2.7
33	−1.78	0.33	33.6	2.6
34	−1.67	0.32	34.4	2.5
35	−1.58	0.31	35.2	2.4
36	−1.49	0.30	35.9	2.4
37	−1.40	0.29	36.6	2.3
38	−1.32	0.28	37.2	2.3
39	−1.24	0.28	37.8	2.2
40	−1.17	0.27	38.4	2.2
41	−1.10	0.27	39.0	2.1
42	−1.03	0.26	39.6	2.1
43	−0.96	0.26	40.1	2.1
44	−0.89	0.26	40.7	2.1
45	−0.82	0.26	41.2	2.1
46	−0.76	0.26	41.7	2.0
47	−0.69	0.25	42.2	2.0
48	−0.63	0.25	42.7	2.0
49	−0.56	0.25	43.3	2.0
50	−0.50	0.25	43.8	2.0
51	−0.44	0.25	44.3	2.0
52	−0.37	0.25	44.8	2.0
53	−0.31	0.26	45.3	2.0
54	−0.24	0.26	45.8	2.0
55	−0.18	0.26	46.4	2.1
56	−0.11	0.26	46.9	2.1
57	−0.04	0.26	47.4	2.1
58	0.03	0.26	48.0	2.1
59	0.10	0.26	48.5	2.1
60	0.17	0.27	49.1	2.1
61	0.24	0.27	49.7	2.2
62	0.31	0.27	50.2	2.2
63	0.39	0.28	50.8	2.2
64	0.47	0.28	51.5	2.2
65	0.54	0.28	52.1	2.3
66	0.63	0.29	52.7	2.3
67	0.71	0.29	53.4	2.3
68	0.80	0.30	54.1	2.4
69	0.89	0.30	54.8	2.4
70	0.98	0.31	55.6	2.5
71	1.08	0.32	56.4	2.5
72	1.18	0.32	57.2	2.6
73	1.29	0.33	58.0	2.6
74	1.40	0.34	58.9	2.7
75	1.52	0.35	59.9	2.8
76	1.65	0.36	60.9	2.9
77	1.78	0.37	61.9	2.9
78	1.92	0.38	63.0	3.0
79	2.07	0.39	64.2	3.1
80	2.22	0.40	65.5	3.2
81	2.39	0.41	66.8	3.3
82	2.57	0.43	68.2	3.4
83	2.76	0.44	69.7	3.5
84	2.96	0.46	71.4	3.7
85	3.19	0.49	73.2	3.9
86	3.44	0.52	75.2	4.1
87	3.73	0.56	77.5	4.5
88	4.09	0.63	80.3	5.0
89	4.56	0.75	84.1	6.0
90	5.31	1.03	90.1	8.2
91	6.56	1.84	100.0	14.7

Same abbreviations as Table 4. SRS-22 logit and percentage measures are quality-of-life measures: the higher the measure, the better the patient. After collection, SRS-22 item scores are rearranged so that the higher the score for each item, the better the quality of life.

## Data Availability

The data presented in this study will be available on Zenodo upon acceptance of the paper.

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
