# Peer review of "Measuring Quality of Life in Adults with Scoliosis: A Cross-Sectional Study Comparing SRS-22 and ISYQOL Questionnaires"

_jcm, 2023, doi:10.3390/jcm12155071_

Round 1

Reviewer 1 Report

Dear authors, congrats for such a huge statistical work/effort. Please find my comments below:

In page 2, lines 90-91 and 94: It would be fine to have a reference here, because there are different sex prevalences described in literature (7:1 above 30º according to SOSORT 2016)

In page 5, about table 1: It is very difficult to understand this table. First of all in the abstract it is said "We recruited 150 adults and 50 adolescents with scoliosis" and here it seems to be infantil, juvenil, etc... Then Mean age (SD) are incongruent values with the groups. Some decimals with points, other with commas... This table needs to be clarified please

It would also be appreciated if the statistical methodology is explained in a more comprehensive manner for those who are not familiarized with it. Maybe not only in appendix but clearly explained in the main text. It would also be fine to see some PCA graphs for loadings for example.

Kindest regards

Reviewer 2 Report

The study compares the construct validity of two questionnaires, ISYQOL and SRS-22, for assessing health-related quality of life (HRQOL) in adults with idiopathic scoliosis. This comparison helps in determining the effectiveness of these questionnaires in measuring disease burden.

Disadvantages:

* The ISYQOL questionnaire, although performing well in adolescent spinal deformities, has not been previously tested in adults. This lack of testing raises uncertainty about its applicability and effectiveness in the adult population with scoliosis.

* The SRS-22 questionnaire showed disordered categories and one item that did not fit the Rasch model. This indicates potential limitations in the structure and item formulation of the questionnaire, which may affect its validity and reliability.

* Both questionnaires demonstrated multidimensionality, suggesting that they measure multiple aspects of HRQOL. This complexity can make interpretation and analysis more challenging and may require additional statistical techniques to handle the multidimensional nature of the data.

* The study found that some items in the questionnaires functioned differently in individuals of different ages. This highlights the potential for response bias and indicates that the questionnaires may not perform consistently across different age groups, which could affect the accuracy and reliability of the results.

The english can be improved. 

Round 2

Reviewer 1 Report

Thanks for your reply letter. I understand your position. Congrats for such a great effort ;)

Reviewer 2 Report

Accept